# The alpha/B.1.1.7 SARS-CoV-2 variant exhibits significantly higher affinity for ACE-2 and requires lower inoculation doses to cause disease in K18-hACE2 mice

**Rafael Bayarri-Olmos[1,2], Laust Bruun Johnsen[3], Manja Idorn[4], Line S Reinert[4], Anne Rosbjerg[1,5], Søren Vang[6], Cecilie Bo Hansen[2], Charlotte Helgstrand[3], Jais Rose Bjelke[3], Theresa Bak-Thomsen[3], Søren R Paludan[4], Peter Garred[2], Mikkel-Ole Skjoedt[2,5]***

[1]Recombinant Protein and Antibody Laboratory, Copenhagen University Hospital, Copenhagen, Denmark; [2]Laboratory of Molecular Medicine, Department of Clinical Immunology, Section 7631, Rigshospitalet Copenhagen University Hospital, Copenhagen, Denmark; [3]Novo Nordisk A/S, Måløv, Denmark; [4]Department of Biomedicine, Aarhus University, Århus, Denmark; [5]Institute of Immunology and Microbiology, University of Copenhagen, Copenhagen, Denmark; [6]Department of Molecular Medicine, Aarhus University Hospital, Aarhus, Denmark

*For correspondence:
moskjoedt@sund.ku.dk

**Competing interest:** The authors declare that no competing interests exist.

**Abstract** The alpha/B.1.1.7 SARS-CoV-2 lineage emerged in autumn 2020 in the United Kingdom and transmitted rapidly until winter 2021 when it was responsible for most new COVID-19 cases in many European countries. The incidence domination was likely due to a fitness advantage that could be driven by the receptor-binding domain (RBD) residue change (N501Y), which also emerged independently in other variants of concern such as the beta/B.1.351 and gamma/P.1 strains. Here, we present a functional characterization of the alpha/B.1.1.7 variant and show an eightfold affinity increase towards human angiotensin-converting enzyme-2 (ACE-2). In accordance with this, transgenic hACE2 mice showed a faster disease progression and severity after infection with a low dose of B.1.1.7, compared to an early 2020 SARS-CoV-2 isolate. When challenged with sera from convalescent individuals or anti-RBD monoclonal antibodies, the N501Y variant showed a minor, but significant elevated evasion potential of ACE-2/RBD antibody neutralization. The data suggest that the single asparagine to tyrosine substitution remarkable rise in affinity may be responsible for the higher transmission rate and severity of the B.1.1.7 variant.

## Introduction

Since the global expansion of SARS-CoV-2, there has been a world-wide surveillance effort to identify new emerging variants empowered by the unprecedented generation of full-genome sequences of circulating strains shared under the GISAID umbrella (http://www.gisaid.org/). Many genetically drifted SARS-CoV-2 variants have been reported since the early spring of 2020. Some of these that affect the S (spike) gene are non-synonymous mutations. These mutations have received much attention since the spike protein binds directly to the angiotensin-converting enzyme-2 (ACE-2) receptor responsible for viral entry into the human host cells (*Hoffmann et al., 2020*; *Zhou et al., 2020*; *Walls et al., 2020*). Furthermore, almost all current vaccine formulations, except for the few vaccines that

rely on the inactivated SARS-CoV-2, focus on raising immunity specifically towards the spike protein or parts of it, mostly the receptor-binding domain (RBD) (*Krammer, 2020*). The possible humoral or cellular immunity evasion due to critical residue changes in B- and T-lymphocyte epitopes of the spike protein is a major concern. In fact, several of the non-synonymous mutations identified to date are located in the RBD of the spike protein and may improve viral fitness by raising the affinity of the virus towards ACE-2 or by hindering antibody-mediated viral neutralization (*Dao et al., 2021*). Some of these residue substitutions, including the N501Y mutation, seem to have arisen independently by convergent evolution and selection and are found in the B.1.1.7 (alpha), the South African B.1.351 (beta), and the Brazilian P.1 (gamma) variants of concern (VOC) (*Rambaut, 2020*; *Faria et al., 2021*; *Tegally et al., 2021*) as well as the variant of interest (VOI) B.1.621 (mu), first identified in Colombia (*Laiton-Donato et al., 2021*), and the former VOI P.3 (theta) from The Philippines (*Tablizo et al., 2021*). All the above VOC/VOI have been for the most part replaced by the B.1.617.2 (delta), which emerged in India in December 2020 (*Cherian et al., 2021*; *Yadav, 2021*) and has become the dominant variant globally (http://www.gisaid.org/, accessed October 2021). A not-yet-peer reviewed publication on vaccine breakthrough, found that the N501Y mutation was associated with increased number of breakthrough infections (*Marques et al., 2021*). Recent studies suggest that the B.1.1.7, B.1.351, P.1, and B.1.617.2 VOC have increased transmissibility (*Faria et al., 2021*; *Tegally et al., 2021*; *Davies et al., 2021a*; *Kidd et al., 2021*; *Scientific Pandemic Influenza Group on Modelling Operational sub-group (SPI-M-O), 2021*) and appear to cause a more severe disease (*Davies et al., 2021b*; *Funk et al., 2021*; *Pearson, 2021*; *Sheikh et al., 2021*), albeit most rely primarily on epidemiological and prediction data. Moreover, other reports indicate that especially the B.1.351, P.1, and B.1.617.2 have a reduced potential of antibody-dependent virus neutralization from convalescent individuals and might pose a challenge to current vaccines (*Wu et al., 2021*; *Madhi et al., 2021*; *Liu et al., 2021b*; *Zhou et al., 2021*; *Dejnirattisai et al., 2021*; *Cele et al., 2021*). The same immune evasion capacity has not been reported for the B.1.1.7 variant, although some evasion cannot be excluded (*Supasa et al., 2021*).

More in-depth knowledge of the functional biophysical characteristics of circulating and emerging variants is needed. A large yeast two-hybrid study characterizing the mutational landscape of SARS-CoV-2 RBD indicated that the N501Y residue-change results in increased affinity towards ACE-2 (*Starr et al., 2020*). Two recent studies, one still non-peer reviewed, have focused on the N501Y mutation showing a variable impact on affinity (from 0.5 nM to sub pM) (*Tian et al., 2021*; *Liu et al., 2021a*).

In the present study, we have performed detailed biophysical characterization of the RBD variants N439K (the most prevalent RBD mutation to date) and N501Y (shared between the B.1.1.7, B.1.351, and P.1), showing that the 1:1 interaction affinity to human ACE-2 is two- and eightfold increased, respectively, compared to the original Wuhan RBD.

Using a transgenic humanized ACE-2 mouse model we found that a lower infection dose was required to establish infection when challenged with the alpha/B.1.1.7 variant compared to an early 2020 isolate (differing at the N501Y position) and that the alpha variant resulted in increased disease severity. Additionally, we present data suggesting a partial decrease in the antibody capacity to neutralize N501Y:ACE-2 interaction in vitro using sera from convalescent individuals (*n* = 140). However, no significant difference was seen when we tested a small cohort of convalescent individuals (*n* = 10) in a plaque reduction neutralization test (PRNT) and a group of anti-RBD monoclonal antibodies (mAbs; *n* = 8).

## Results

### Biophysical characterization

We expressed recombinant SARS-CoV-2 RBD wild-type (wt) (Wuhan-hu-1), RBD N439K, and RBD N501Y in Expi293 HEK cells and performed thermal stability and binding kinetic analyses to determine the biophysical relevance of the RBD variants (*Figure 1*). Protein purity and homogeneity were evaluated by sodium dodecyl sulphate–polyacrylamide gel electrophoresis (SDS–PAGE; *Figure 1A*) and size-exclusion high-performance liquid chromatography (HPLC; *Figure 1B*). We monitored the thermal unfolding using the intrinsic fluorescence ratio at 350 and 330 nm and observed a ~2.5°C reduction in the inflection temperature (Ti) for the N439K variant (*Figure 1C*). This suggests that the N439K, but not the N501Y, has a moderately deleterious effect on the RBD stability. Next, we

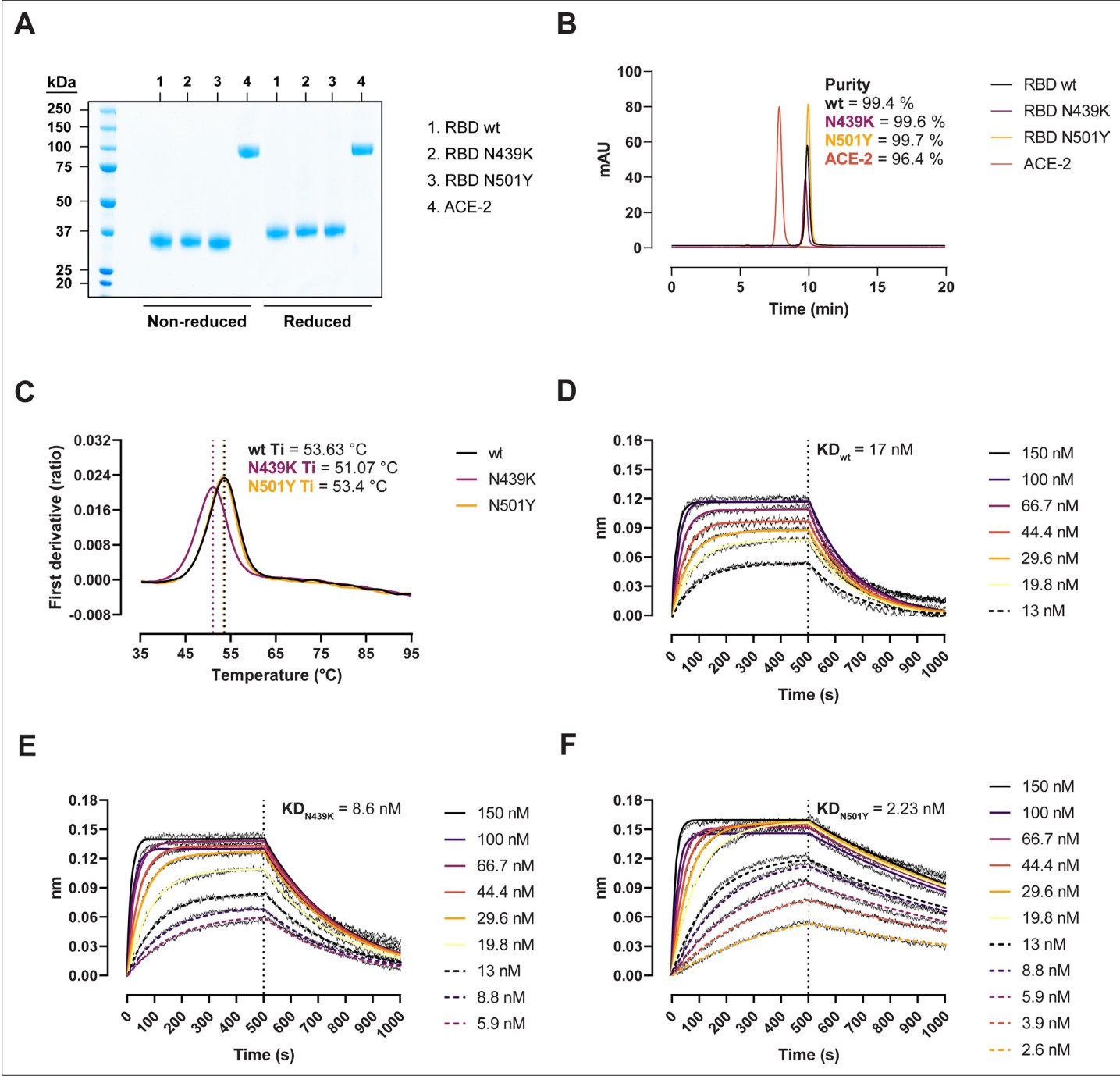

**Figure 1.** Biophysical characterization of recombinant receptor-binding domain (RBD) variants. (**A**) Sodium dodecyl sulphate–polyacrylamide gel electrophoresis (SDS–PAGE) total protein stain of RBD variants and angiotensin-converting enzyme-2 (ACE-2). (**B**) Size-exclusion chromatography (SEC) profiles of the purified proteins run in a BioSep-SEC-S3000 column. Purity was determined by peak integration with the Empower software. (**C**) Thermal denaturation curves of the RBD wild-type (wt), N439K, and N501Y variants. Data are represented as the first derivative of the intrinsic fluorescence ratio 350:330 nm of the mean of three replicates. Vertical dashed lines represent the inflection temperatures (Ti). Biolayer interferometry (BLI) sensorgrams of RBD wt (**D**), N439K (**E**), and N501Y (**F**) binding to ACE-2-Fc immobilized in anti-human Fc capture (AHC) sensors. ACE-2-immobilized sensors were dipped into 7- to 11-point dilution series of RBD for 500 s, followed by dissociation for another 500 s.

The online version of this article includes the following figure supplement(s) for figure 1:

**Source data 1.** Accompanying *Figure 1*.

**Source data 2.** Accompanying *Figure 1*.

**Source data 3.** Accompanying *Figure 1*.

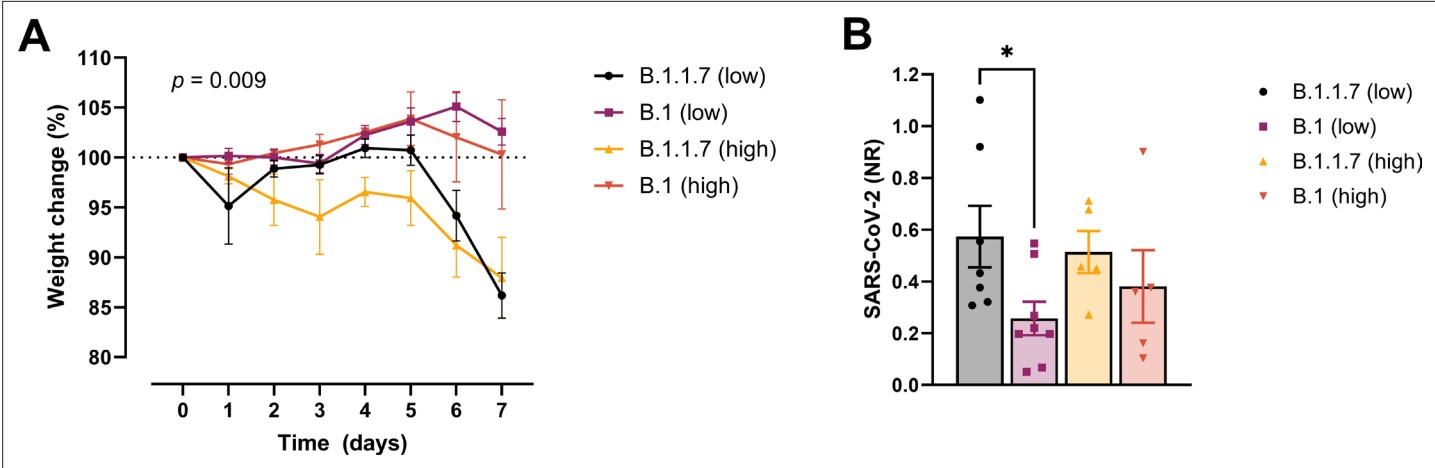

**Figure 2.** Development of COVID-19-like disease in mice. (**A**) Weight evolution of K18-hACE2 mice challenged intranasally with $2.5 \times 10^3$ (low, $n = 9$ per group) or $2.5 \times 10^4$ p.f.u. (high, $n = 5$ per group) SARS-CoV-2 B.1 (Freiburg isolate, FR-4286) or B.1.1.7. Global differences between the groups were analysed with the Kruskal–Wallis test. (**B**) Viral load measured in the lungs on day 2 post-infection. The mice were infected with the same doses as described in (**A**). The expression of SARS-CoV-2 RNA was analysed by real-time quantitative polymerase chain reaction (qPCR). Values normalized to 18S-rRNA are presented as mean ± standard error of the mean (SEM). * = 0.019.

The online version of this article includes the following figure supplement(s) for figure 2:

**Source data 1.** Accompanying *Figure 2*.

measured their binding kinetics towards the human ACE-2 by biolayer interferometry (BLI) to determine their functional importance. The N439K variant bound with an approx. twofold higher affinity than the wt (8.6 vs 17 nM) (*Figure 1D, E*), while the N501Y variant did so with an eightfold higher affinity (2.23 vs 17 nM) (*Figure 1F*). Analyses of the binding response curves indicate that the variants bound to ACE-2 faster ($Ka_{N439K} = 4.15 \times 10^5$ $M^{-1}$ $s^{-1}$, $Ka_{N501Y} = 4.76 \times 10^5$ $M^{-1}$ $s^{-1}$, $Ka_{wt} = 3.34 \times 10^5$ $M^{-1}$ $s^{-1}$) and mostly the N501Y had noticeable slower dissociation rates ($Kdis_{N439K} = 3.57 \times 10^{-3}$ $s^{-1}$, $Kdis_{N501Y} = 1.06 \times 10^{-3}$ $s^{-1}$, $Kdis_{wt} = 5.9 \times 10^{-3}$ $s^{-1}$).

## Alpha/B.1.1.7 establishes disease-causing infection at lower inoculation doses than original SARS-CoV-2 isolate

In order to examine whether the increased affinity of the N501Y variant for ACE-2 was associated with a more efficient establishment of infection and development of disease, we challenged transgenic ACE-2 humanized K18-hACE2 mice with the early 2020 SARS-CoV2 B.1 (Freiburg isolate, FR-4248 *Miladi, 2020*) and the B1.1.7 (alpha) strains. The model has been reported to reflect many aspects of COVID-19, including viral replication and histopathological changes in the lungs (*Winkler et al., 2020*). Upon infection with two different doses of the B.1 strain, we observed no weight loss as an indication of disease development (*Figure 2A*). However, infection with B.1.1.7 at the same doses led to severe disease development in the mice. Mann–Whitney pair-wise comparisons (B.1 vs B.1.1.7) showed a significant difference in weight at low viral doses at days 6 and 7 (multiple comparisons corrected $q$ values 0.009743 and 0.000291, respectively, $n = 9$ per group). The same tendency, albeit not statistically significant, was observed for the high viral doses ($n = 5$ per group). We also observed that the virus replicates to higher levels in the lungs at day 2 post-infection with B.1.1.7 compared to B.1 when infected with low viral doses (*Figure 2B*).

## Determination of the evasion capacity of the N439K and N501Y variants in naturally induced antibody-mediated immunity

Next, we sought to clarify whether the residue changes could affect the folding and binding properties of the RBD and viral fitness by providing immune evasion. To do so, we first determined the antibody-mediated inhibition potency, measured as the inhibition of the ACE-2/RBD interaction, in sera of recovered COVID-19 patients ($n = 140$) using a validated in vitro antibody inhibition assay (*Bayarri-Olmos et al., 2021a*; *Figure 3A–C*). There was a statistically significant reduction in the

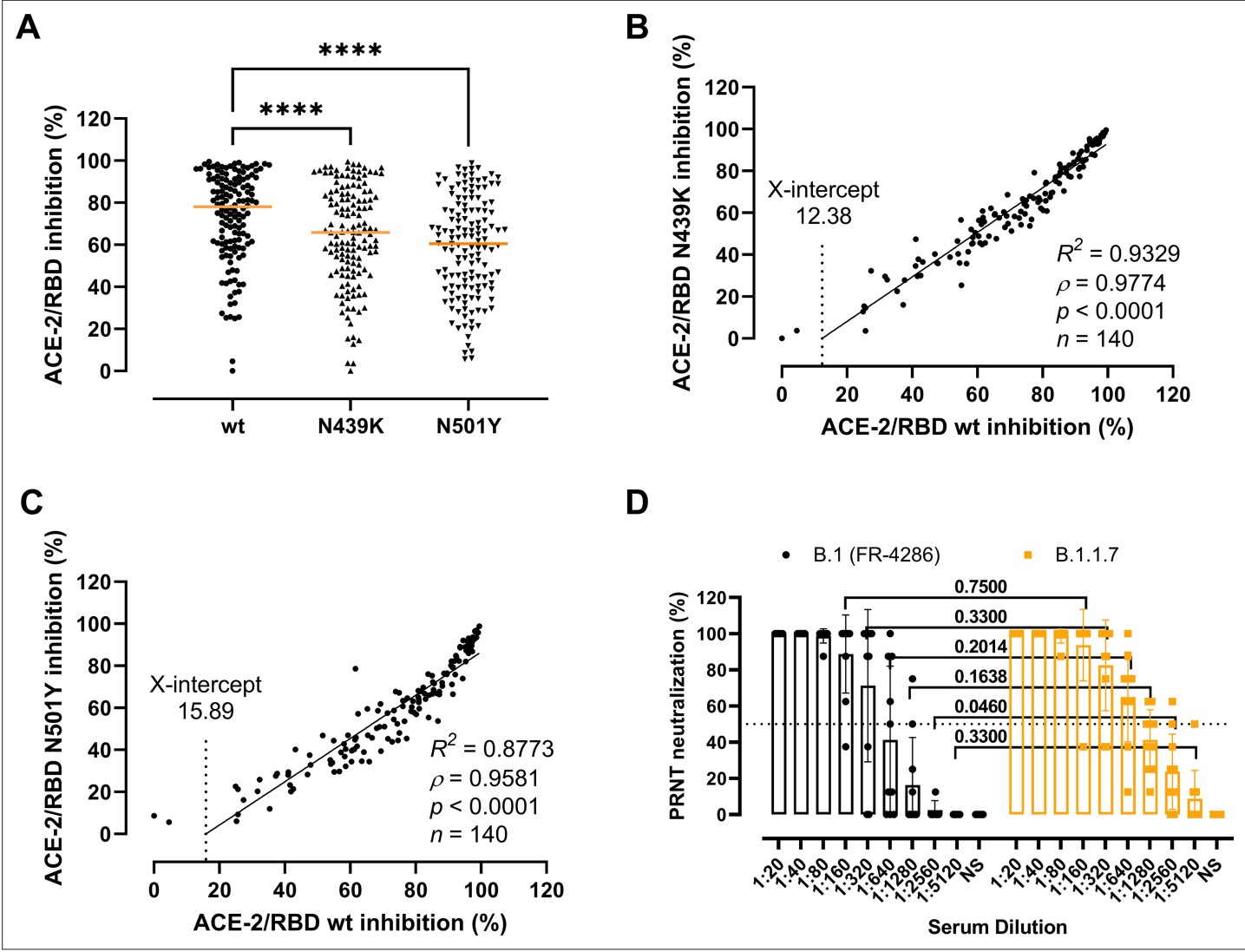

**Figure 3.** Antibody-mediated inhibition potency of recovered COVID-19 patient sera. (**A**) Inhibition of wild-type (wt), N439K, and N501Y receptor-binding domain (RBD) towards angiotensin-converting enzyme-2 (ACE-2) in serum from convalescent COVID-19 individuals ($n = 140$). Friedman test with Dunn's multiple comparisons. Orange lines represent medians. ****$p < 0.0001$. Linear regression and Spearman correlation analyses for N439K vs wt (**B**) and N501Y vs wt (**C**). Trend line represents linear regression. (**D**) Neutralization of serum from convalescent COVID-19 individuals ($n = 10$) against B.1 and B.1.1.7 calculated by the plaque reduction neutralization test (PRNT) and analysed using Wilcoxon matched-pairs signed ranked tests with multiple comparisons corrections. NS, no serum.

The online version of this article includes the following figure supplement(s) for figure 3:

**Source data 1.** Accompanying *Figure 3*.

inhibition of the N439K and N501Y RBD compared to the wt ($p < 0.0001$ for both) (*Figure 3A*). The inhibition potencies towards the wt and the variants had a highly significant correlation ($\rho = 0.9774$ and $\rho = 0.9581$ for N439K and N501Y, respectively, $p < 0.0001$), with the best-fit X-intercept ranging from 12.38 to 15.89 for the N439K and N501Y, respectively (*Figure 3B, C*). However, analyses of convalescent sera ($n = 10$) using a PRNT virus neutralization platform (*Fougeroux et al., 2021*) showed no significant difference in the neutralization potency towards the B.1 and B.1.1.7 strains (*Figure 3D*).

## Determination of the evasion capacity of the RBD variants in vaccine-induced immunity

Next, we analysed the variants' evasion capacity using a previously established vaccine mouse model (*Sheikh et al., 2021*). Briefly, mice were immunized with wt RBD ($n = 3$) or wt prefusion-stabilized

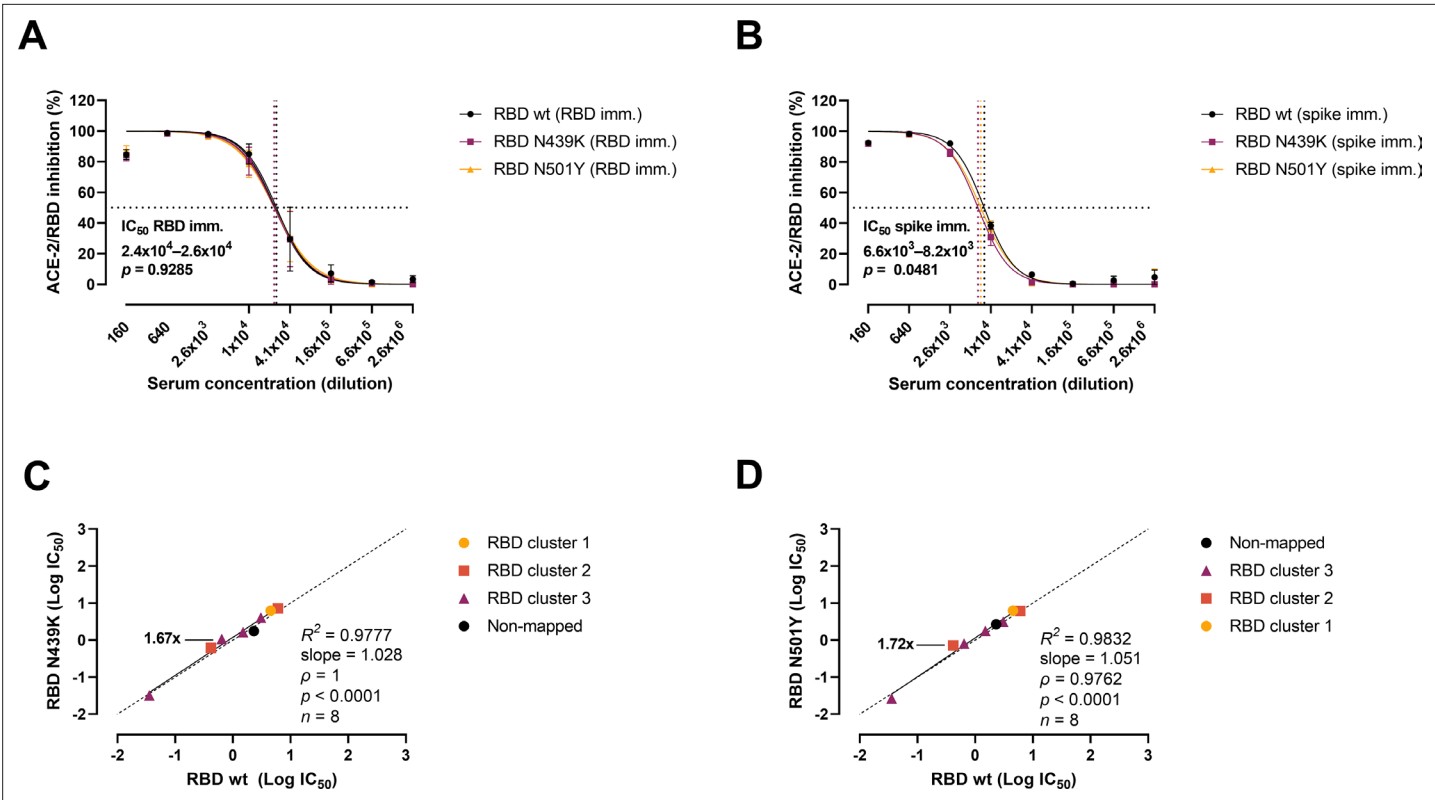

**Figure 4.** Antibody-mediated inhibition potency of polyclonal sera and monoclonal antibodies (mAbs) isolated from mice immunized against wild-type (wt) receptor-binding domain (RBD) or spike protein. $IC_{50}$ comparison of the inhibition of polyclonal mice sera from an animal vaccine model based on RBD (**A**) or spike (**B**) challenged with RBD wt, N439K, and N501Y. Connecting lines represent non-linear fits using the equation [inhibitor] vs normalized response with variable slope. Vertical dashed lines delimit the $IC_{50}$ values, where they intersect the horizontal 50% inhibition dashed line. Data are presented as mean ± standard error of the mean (SEM). Comparison of the inhibition potency (log[$IC_{50}$]) of mouse mAbs ($n = 8$) calculated from three independent experiments against RBD wt and RBD N439K (**C**) or RBD N501Y (**D**), analysed by linear regression and Spearman correlation. The trend represents a linear regression. Dashed line signals equidistance between axis (i.e. slope 1). Neutralization fold changes over 1.5 are highlighted. Only mAbs with $IC_{50}$ values within the range of concentrations tested were included in the statistical analyses (i.e. 8 out of 18 tested mAbs).

The online version of this article includes the following figure supplement(s) for figure 4:

**Source data 1.** Accompanying *Figure 4*.

spike protein ectodomain ($n = 3$). Polyclonal sera were collected after three rounds of immunizations, and mAbs were developed and characterized from cloned hybridomas. As shown previously (***Bayarri-Olmos et al., 2021a***; ***Bayarri-Olmos et al., 2021b***), polyclonal sera from RBD-immunized mice were approx. fourfold more effective than spike-immunized mice sera, with $IC_{50}$ values of $2.4–2.6 \times 10^4$ (RBD) and $6.6–8.2 \times 10^3$ (spike), respectively (***Figure 4A, B***). Sera from RBD-immunized mice showed no difference in the inhibitory potency against the wt and the variants. However, best-fit $IC_{50}$ values from sera from mice immunized with spike differed slightly between strains, as 11% and 19% higher serum concentration was required for the N439K and N501Y variants, respectively, to achieve the same inhibition levels as for the wt RBD. We also evaluated the inhibition potency of mouse mAbs raised against wt RBD and wt spike ($n = 18$) (***Figure 4C, D***). The mAbs were screened for high affinity towards the SARS-CoV-2 RBD wt, and their epitopes mapped via competition assays (***Bayarri-Olmos et al., 2021a***). Linear regression and Spearman correlation analyses of the mAbs with best-fit $IC_{50}$s within the range of concentration tested ($n = 8$) showed that the N439K and N501Y mutations had very minor effects on the inhibition potency of the mAbs (N439K $R^2 = 0.9777$, $\rho = 1$, p < 0.0001; N501Y $R^2 = 0.9832$, $\rho = 0.9762$, p < 0.0001).

## Discussion

Emerging clusters of genetically drifted SARS-CoV-2 variants have received much attention due to concerns of enhanced adaptive fitness and viral escape of neutralizing antibodies or T-cell-mediated responses. A specific focus has been on the spike protein changes that interact with the ACE-2 receptor and mediate host cell entry. This is also the target for the vast majority of the vaccine strategies. A worldwide effort to sequence viral strains has revealed many emerging variants. However, although many spike variants have been reported so far, there seems to be a limitation in terms of the 'freedom' possibilities to which and how many non-synonymous mutations are emerging in the S gene and particularly in the RBD coding region (*Starr et al., 2020*). One of these RBD residue changes is N501Y, which has appeared by convergent evolution in three of the so-called VOC: B.1.351 (beta), P.1 (gamma), and the B.1.1.7 (alpha) variant, but interestingly not in the rapidly spreading B.1.617.2 (delta) variant, where other mutations appear to be of importance. Nevertheless, from an epidemiological viewpoint, all these variants seem to have a higher transmission rate and have outcompeted the original Wuhan strain in the regions they have arisen or been introduced to. In Europe, the B.1.1.7 (alpha) variant was, at the end of March 2021, dominating many countries only few months after its emergence with severe consequences on the health care system, while now just a few months later in the end of August 2021, the B1.617.2 (delta) variant has taken over (http://www.gisaid.com/, accessed August 2021).

We aimed to characterize the functional properties of the B.1.1.7 (alpha) variant and address any potential evasion capacity of antibody-mediated neutralization. We also did this for the prevalent RBD mutation N439K and we found a twofold affinity increase to ACE-2 and a partial evasion of antibody-mediated neutralization, lending support to a recently published paper (*Dao et al., 2021*).

When we did BLI measurements of the 1:1 interaction of N501Y RBD on ACE-2 immobilized sensors, we found a remarkable eightfold affinity increase of the variant (2.2 nM) compared to wt Wuhan RBD (17 nM). This might be a central explanation for the observed higher transmission rate and the fitness advantage of the B.1.1.7 variant. Furthermore, the 1:1 molecular affinity determination might even underestimate the true in vivo interaction potential covering multivalent avidity interactions between the trimeric spike scaffolds on the viral surface and the ACE-2-covered host membrane. In agreement with this, we found that transgenic ACE-2 humanized K18-hACE2 mice developed severe disease following a low inoculation dose of B.1.1.7, which did not cause disease for the early B.1 strain. Another report has shown that the B.1.1.7 variant might also infect WT mice though not as efficient as the P.1 and B.1.351 (*Tian et al., 2021*).

Two recent studies, one yet-to-be peer reviewed, have focused on the biophysics of the N501Y variant, reporting affinities ranging from 0.5 nM (*Tian et al., 2021*) to undefined sub pM (*Liu et al., 2021a*). Even though they numerically deviate from our findings, they conceptually observe increased affinity of the N501Y variant. However, in both studies, immobilized RBD was incubated with dimeric ACE-2 in the soluble phase. Therefore, avidity interactions might have contributed to the determined $K_D$ values using the 1:1 fitting model employed in the studies. Of note, our affinity findings are in good agreement with those of a recent study using surface plasmon resonance spectrometry to evaluate the binding between immobilized ACE-2 and soluble RBD ($K_D$ 2.4 ± 0.4), published during the revision process for the current manuscript (*Laffeber et al., 2021*).

Whether the alpha variant escapes immunity is still a matter of debate, but it has been shown that alpha might confer some reduction in virus neutralization potential compared with the wt both in naturally infected and vaccinated individuals (*Supasa et al., 2021*). However, the decrease in response varies highly between individuals and whether the variation has clinical relevance requires further studies.

When we tested the antibody-mediated inhibition of the wt, N439K, and N501Y RBD interaction with human ACE-2, we found a minor, but significant reduction of the neutralization potency of convalescent sera ($n$ = 140) compatible with the data observed by *Supasa et al., 2021*. The relative neutralization evasion level from these experiments was WT RBD < N439K < N501Y. Conversely, hyperimmune sera and high-affinity mAbs from mice that received several immunizations with recombinant wt spike or RBD did not show this significant neutralization difference, indicating that a fully established vaccine response will overcome the evasion potential of the N501Y. As shown by others (*Ravichandran et al., 2020*) and us *Bayarri-Olmos et al., 2021a*, focused immunization with RBD leads to higher antibody titres and higher neutralization potency than immunization with the full spike

ectodomain. At sufficiently high antibody titres, the N439K and N501Y variants' potential evasion advantage might of less immunological importance.

When we addressed the antibody neutralization in sera from a group of convalescent individuals using the PRNT assay, we found no statistical difference between the early 2020 Freiburg isolate and the alpha variant; an observation also made by others performing live virus PRNT assays (*Liu et al., 2021b*), which is not in full agreement with the study from *Supasa et al., 2021*. The discrepancy between the findings can be attributed to several reasons: First, the viral neutralization assays are challenging to perform and might be hampered by intrinsic high biological variation compared with the ACE-2/RBD ELISA inhibition assay that can be performed on a large scale and is optimized to display little assay variability. Another critical issue is that the neutralization assays address the whole virus and contain multiple elements involved in the interaction and infection, that is membrane clusters of trimeric spike and the intracellular structural proteins that determine viral fitness, which might lead to differences. Since the different neutralization studies involve relatively few investigated individuals, the risk of type 1 and type 2 errors should be taken into account. Interestingly, one study proposes that the N501Y variant might pose challenges for the MHC class II presentation and the CD4+ T-cell response (*Castro et al., 2021*). In contrast to this, another study has demonstrated a negligible impact of SARS-CoV-2 variants on CD4+ and CD8+ T-cell reactivity (*Tarke et al., 2021*).

In this study, we present comprehensive biophysical data showing that the single N501Y residue change in the RBD region results in an eightfold increase in the affinity to human ACE-2. This affinity adaptation is likely also to be found in the two other VOC, the B.1.351 and P.1. In line with this, we find that the alpha variant induces a more severe disease that an early 2020 SARS-CoV-2 isolate in K18-hACE2 mice, indicating a more efficient establishment of infection in vivo. Moreover, our data show a minor but significant immune evasion effect of the N501Y substitution.

## Materials and methods

### Production of recombinant SARS-CoV-2 RBD variants and human ACE-2 ectodomain

The nucleotide sequence corresponding to the SARS-CoV-2 RBD (QIC53204.1, aa R319–S593) with either an N439K or N501Y substitution and a C-terminal 10xHis-AviTag were synthesized and subcloned into pcDNA3.4-TOPO expression vectors by GeneArt (Thermo Fisher Scientific, Massachusetts, USA). The sequences were optimized towards higher codon adaptation indexes, 5′ mRNA folding energies, removal of cryptic splice sites, and tandem repeats as described elsewhere (*Hertz et al., 2018*). The production and purification of recombinant human ACE-2 ectodomain and ACE-2-Fc; and production, purification, and biotinylation of the RBD variants were performed as described elsewhere (*Bayarri-Olmos et al., 2021a*; *Bayarri-Olmos et al., 2021b*). Protein purity and quality were determined by 4–12% Bis-Tris SDS–PAGE and Coomassie staining and size-exclusion HPLC. The recombinantly produced RBDs were identified as having the expected mass by intact mass LC–MS.

### Protein stability determination

The impact of the N439K and N501Y substitutions on the thermal stability of the RBD was analysed in triplicates on a Tycho NT.6 (NanoTemper Technologies GmbH, Munich, Germany) using a thermal ramp of 30°C/min in phosphate-buffered Saline (PBS). Protein unfolding was assessed from the ratio of the intrinsic fluorescence recorded at 350 and 330 nm. Inflection temperatures (Ti), representing a discrete unfolding transition or change in the structural integrity of the proteins, were calculated by the Tycho software.

### ACE-2/RBD affinity determination by BLI

Binding kinetics experiments were performed on an Octet RED383 system (ForteBio, California, USA) as described elsewhere with minor modifications (*Bayarri-Olmos et al., 2021b*). Briefly, 13 µg/ml ACE-2-Fc was loaded on anti-human Fc capture sensors (Pall Life Sciences, California, USA) for 500 s, followed by baseline for 60 s, association to 12-point 1.5-fold serial dilutions starting at 150 nM for RBD wt, N439K, and N501Y for 500 s, and finally dissociation for another 500 s. For each 16-channel column sensor, four sensors loaded with ACE-2-Fc were assigned as a reference and dipped into

buffer during the association and dissociation phases. Final sensorgrams were reference subtracted column-wise and globally fitted to a 1:1 binding model.

## ACE-2/RBD antibody inhibition assay

The antibody neutralization potency, calculated from the degree of inhibition of the ACE-2:RBD interaction, was measured in serum samples from recovered individuals with a past PCR-confirmed COVID-19 infection, mice immunized with wt RBD or spike, and anti-RBD mouse mAbs using an ELISA-based ACE-2/RBD antibody inhibition test described elsewhere (*Bayarri-Olmos et al., 2021a*). Briefly, ACE-2 ectodomain (1 μg/ml) was coated onto Nunc Maxisorp microtitre plates (Thermo Fisher Scientific) overnight at 4°C in PBS. Samples were incubated with a solution of biotinylated RBD (4 ng/ml) and Pierce high sensitivity streptavidin (Thermo Fisher Scientific) (1:16,000) as follows: convalescent sera in a 10% dilution, immunized mice sera in an eight-point fourfold dilution starting at a 0.625% dilution, and mAbs in a six-point fourfold dilution starting at 20 μg/ml. After 1 hr incubation in low-binding round-bottom plates (Thermo Fisher Scientific), the samples/RBD mixes were transferred to ACE-2-coated wells for 15 min. The plates were developed with TMB One (KemEnTec Diagnostics, Taastrup, Denmark) for 20 min, the reaction stopped with 0.3 M $H_2SO_4$, and the optical density recorded at 450 nm. Coating buffer (PBS, 10.1 mM $Na_2HPO_4$, 1.5 mM $KH_2PO_4$, 2.7 mM KCl, 137 mM NaCl), sample and washing buffer (PBS, 0.05% Tween-20). The wells were washed three times with washing buffer between steps, and all incubations—unless otherwise stated—were done at room temperature in an orbital shaker.

## K18-hACE2 mouse COVID-19 model

K18-hACE C57BL/6J mice (strain: 2B6.Cg-Tg(K18-ACE2)2Prlmn/J) were obtained from The Jackson Laboratory (Stock number: 034860). Age-matched male and female mice, randomized in groups, were fed standard chow diet and housed in a pathogen-free facility. Animals were anesthetized with an intraperitoneal injection of ketamine (100 mg/kg body weight) and xylazine (10 mg/kg body weight) and administered either $2.5 \times 10^3$ or $2.5 \times 10^4$ plaque-forming units (p.f.u.) SARS-CoV-2 via intranasal administration. Mice were weighed every day at the same time of the day until day 7 post-infection.

## RNA isolation and real-time PCR (qPCR)

Lungs were homogenized with steel beads in a Tissuelyser (II) (both from Qiagen, Hilden, Germany) in PBS and immediately used for RNA isolation. RNA was isolated using the High Pure RNA Isolation Kit (Roche, Basel, Switzerland) and an equal amount of RNA was used for standard One-Step RT-PCR (TaqMan RNA-to-Ct 1-Step Kit, Applied Biosystems, Massachusetts, USA). For the SARS-CoV-2 *N* gene, qPCR primers AAATTTTGGGGACCAGGAAC and TGGCACCTGTGTAGGTCAAC and probe FAM-ATGTCGCGCATTGGCATGGA-BHQ were used. For the 18S-rRNA qPCR the Hs99999901_s1 18S rRNA Taqman Gene Expression Assays were used (Applied Biosystems). RNA levels of SARS-CoV-2 *N* gene were normalized to the mouse housekeeping gene 18S-rRNA using the formula: 2Ct(18S-rRNA) − Ct(Sars-CoV-2 RNA). The resulting normalized ratio is presented directly in the figures.

## SARS-CoV-2

The early 2020 B.1 strain, Freiburg isolate FR-4286, was kindly provided by Professor Georg Kochs, University of Freiburg, Germany and B.1.1.7 SARS-CoV2 (Kent, UK, isolate) was provided under MTA by Professor Arvind Patel, University of Glasgow. The viruses were propagated in VeroE6 cells expressing human TMPRSS2 (VeroE6-hTMPRSS2) (kindly provided by Professor Stefan Pöhlmann, University of Göttingen) (*Hoffmann et al., 2020*). We have confirmed the expression of hTMPRSS2 in the Green monkey VeroE6 cell line with PCR analysis and the cells were tested negative for mycoplasma contamination. Briefly, VeroE6-hTMPRSS2 cells were infected with a multiplicity of infection of 0.05, in Dulbecco's modified Eagle Medium (DMEM; Gibco) + 2% Fetal calf serum (FCS) (Sigma-Aldrich) + 1% Pen/Strep (Gibco)+ L-glutamine (Sigma-Aldrich) (from here, complete medium). The supernatant containing new virus progeny was harvested 72 hr post-infection, and concentrated on 100 kDa Amicon ultrafiltration columns (Merck, New Jersey, USA) by centrifugation at 4000 × *g* for 30 min. Virus titer was determined by $TCID_{50\%}$ assay and calculated by the Reed–Muench method (*Reed and Muench, 1938*).

Virus isolates were prepared for whole-genome sequencing using the EasySeq RC-PCR SARS CoV-2 Whole Genome Sequencing kit (Nimagen, Nijmegen, Netherlands) and sequenced using Illumina sequencing (Illumina, California, USA). Adapter and primer sequences were removed, and reads were mapped to the NC_045512.2 reference using Minimap and iVar (PMID: 29750242, PMID: 30621750). Variants and pangolin lineages (PMID: 32669681) were determined based on iVar consensus sequences with minimum 80% base frequency.

The Freiburg isolate FR-4286 is the Wuhan-like early European B.1 lineage containing the S:D614G, and ORF1b:P314L, as well as E:L37R. The B.1.1.7 differs from the B.1 in the spike protein on positions S:N501Y, S:A570D, S:T716I, S:P681H, S:S982A, S:D1118H, S:del69/70, and S:del144/145. Both the B.1.1.7 and the B.1 strains contains the S:D614G and ORF1b:P314L mutations. The viruses used are clinical isolates. The B.1.1.7 variant is in the database as MZ314997. The Freiburg isolate sequence has been uploaded and has the accession ID no: EPI_ISL_852748.

## Plaque reduction neutralization test

Neutralizing capacity of convalescent serum against B.1 SARS-CoV2 (FR-4286) or B.1.1.7 was assessed in a neutralization assay (PRNT), performed as previously described (*Fougeroux et al., 2021*). In short, convalescent serum from human COVID-19 patients was heat inactivated (30 min at 56°C) and prepared in a nine-point twofold serum dilution starting at 1:20, in DMEM (Gibco) + 2% FCS (Sigma-Aldrich) + 1% Pen/Strep (Gibco) + L-glutamine (Sigma-Aldrich). Sera were mixed with SARS-CoV-2 to a final titre of 100 $TCID_{50\%}$/well, and incubated at 4°C overnight. A 'no serum' and a 'no virus' (uninfected) control samples were included. $TCID_{50\%}$ control plates (in triplicates) of each of the viruses were included to control for actual virus titre of B.1 and B.1.1.7, respectively. The following day, virus:serum mixtures were added in octuplicates to $2 \times 10^4$ VeroE6-hTMPRSS2 cells (*Hoffmann et al., 2020*) seeded in flat-bottomed 96-well plates (Thermo Fisher), and incubated 72 hr in a humidified $CO_2$ incubator at 37°C, 5% $CO_2$. Cytopathic effect was scored after fixing with 5% formalin (Sigma-Aldrich) and crystal violet stain (Sigma-Aldrich), using a light microscope (Leica DMi1).

## Blood samples

The antibody neutralization potency was assessed in 140 randomly selected convalescent serum samples (the patient cohort has been described elsewhere *Hansen et al., 2021*) with $IC_{50}$ values for the ACE-2/RBD wt interaction ranging from low to high (estimated from six-point fourfold dilution series done as part of a previous study *Bayarri-Olmos et al., 2021a*). A serum pool from healthy individuals was used as a negative control.

## Biosafety

All aspects of this study were approved by the office of the Danish Working Environment Authority, Landskronagade 33, 2,100 Copenhagen Ø, before the initiation of this study. Work with SARS-CoV-2 was performed in a biosafety level 2+ laboratory by personnel equipped with powered air-purifying respirators.

## Statistics

Statistical analyses were performed with GraphPad Prism 9 (GraphPad Software, California, USA). Global differences in the weight of K18-hACE2 mice exposed to high and low doses of SARS-CoV-2 were analysed with Kruskal–Wallis. Pair-wise comparisons of the effect of SARS-CoV-2 variants in weight loss were performed with multiple Mann–Whitney tests (ranks computed for each day) and false discovery rate approach using the two-stage step-up method of Benjamini, Krieger, and Yekutieli. Comparison of the neutralization potency of serum samples from COVID-19 recovered, vaccinated individuals, and mouse mAbs was performed by Friedman test comparing the mean of each of the variants with the wt, as well as pair-wise (variant vs wt) Spearman rank correlation tests (two-tailed, reported as $\rho$ and a significance value $p$) and linear regression analyses (reported as $R^2$). Neutralization indexes of convalescent sera for the B.1 and B.1.1.7 strains obtained from the PRNT were analysed by Wilcoxon matched-pairs signed ranked tests corrected for multiple comparisons using the Holm–Šídák method. The extra-sum-of-squares $F$-test was used to compare best-fit $IC_{50}$ values, interpolated with the equation [inhibitor] vs normalized response with a variable slope as described elsewhere (*Sheikh et al., 2021*), of mice sera and mice mAbs. p values <0.05 were considered statistically significant.

## Acknowledgements

The authors would like to thank Camilla Xenia Holtermann Jahn, Sif Kaas Nielsen, Bettina Eide Holm, and Mads Engelhardt Knudsen from the Laboratory of Molecular Medicine at Rigshospitalet, and Ea Stoltze Andersen from the Department of Molecular Medicine at Aarhus Universy Hospital for their excellent technical assistance, and Per Franklin Nielsen from Novo Nordisk A/S for mass spectrometry identification of all recombinant proteins. Professor Stefan Pöhlman, University of Göttingen, Germany kindly provided us with VeroE6-hTMPRSS2 cells, and Professors Georg Kochs, University of Freiburg and Arvind Patel, University of Glasgow, United Kingdom, kindly provided us with the SARS-CoV2 Freiburg isolate and the alpha/B.1.1.7 SARS-CoV-2 Kent isolate, respectively. This work was financially supported from grants from the Carlsberg Foundation (CF20-0045), independent Research Fund Denmark (0214-00001B), The European Research Council (ERC-AdG ENVISION; 786602), and the Novo Nordisk Foundation (NFF205A0063505, NNF20OC0063436, and NNF20SA0064201).

## Additional information

### Funding

| Funder | Grant reference number | Author |
|---|---|---|
| Carlsbergfondet | CF20-0045 | Rafael Bayarri-Olmos<br>Anne Rosbjerg<br>Peter Garred<br>Mikkel-Ole Skjoedt |
| Novo Nordisk Fonden | NFF205A0063505 | Rafael Bayarri-Olmos<br>Anne Rosbjerg<br>Peter Garred<br>Mikkel-Ole Skjoedt |
| Novo Nordisk Fonden | NNF20OC0063436 | Rafael Bayarri-Olmos<br>Anne Rosbjerg<br>Peter Garred<br>Mikkel-Ole Skjoedt |
| Novo Nordisk Fonden | NNF20SA0064201 | Rafael Bayarri-Olmos<br>Anne Rosbjerg<br>Peter Garred<br>Mikkel-Ole Skjoedt |
| European Research Council | ERC-AdG ENVISION; 786602 | Søren R Paludan |
| Independent Research Fund Denmark | 0214-00001B | Søren R Paludan |

The funders had no role in study design, data collection, and interpretation, or the decision to submit the work for publication.

### Author contributions

Rafael Bayarri-Olmos, Conceptualization, Data curation, Formal analysis, Investigation, Methodology, Writing - original draft, Writing – review and editing; Laust Bruun Johnsen, Data curation, Formal analysis, Investigation, Methodology, Writing – review and editing; Manja Idorn, Formal analysis, Methodology, Writing – review and editing; Line S Reinert, Anne Rosbjerg, Investigation, Methodology, Writing – review and editing; Søren Vang, Formal analysis, Investigation, Methodology; Cecilie Bo Hansen, Data curation, Formal analysis, Writing – review and editing; Charlotte Helgstrand, Jais Rose Bjelke, Data curation, Investigation, Methodology, Writing – review and editing; Theresa Bak-Thomsen, Conceptualization, Project administration, Writing – review and editing; Søren R Paludan, Conceptualization, Funding acquisition, Project administration, Writing – review and editing; Peter Garred, Conceptualization, Project administration, Writing - original draft, Writing – review and editing; Mikkel-Ole Skjoedt, Conceptualization, Data curation, Formal analysis, Funding acquisition, Investigation, Methodology, Project administration, Supervision, Validation, Writing - original draft, Writing – review and editing

### Author ORCIDs

Peter Garred http://orcid.org/0000-0002-2876-8586
Mikkel-Ole Skjoedt http://orcid.org/0000-0003-1306-6482

### Ethics

The collection and use of blood samples has been approved by the Regional Ethical Committee of the Capital Region of Denmark (H-20028627 and H-20079890). The human studies were conducted in agreement with the Helsinki declaration. We have received informed consent to do the examinations included in this study including to publish data.

The Danish Animal Experiments Inspectorate has approved the experimental animal procedures and they were carried out in accordance with the Danish Animal Welfare Act for the Care and Use of Animals for Scientific Purposes (License ID 2019-15-0201-00090 and 2020-15-0201-00726). All procedures followed the recommendations of the Animal Facilities at the Universities of Copenhagen and Aarhus.

### Decision letter and Author response

Decision letter https://doi.org/10.7554/eLife.70002.sa1
Author response https://doi.org/10.7554/eLife.70002.sa2

## Additional files

### Supplementary files

• Transparent reporting form

### Data availability

All data generated or analysed during this study are included in the manuscript and supporting files. We have no restrictions with regards to data availability.

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
