## [Decision Letter]

**Decision letter after peer review:**

Thank you for submitting your article "The B.1.1.7 SARS-CoV-2 variant exhibits significantly higher affinity for ACE-2 and accelerates disease severity in transgenic hACE-2 mice" for consideration by *eLife*. Your article has been reviewed by 3 peer reviewers, and the evaluation has been overseen by a Reviewing Editor and Miles Davenport as the Senior Editor. The reviewers have opted to remain anonymous.

Essential revisions:

1) To assess the pathogenicity and potential immune evasion of B.1.1.7, the authors use a control isolate termed FR-4286. In which sites do the Spike proteins of B.1.1.7 and FR-4286 differ? The respective sequences may be provided. Can the authors exclude mutations that occurred during propagation in cell culture (in Vero cells)?

2) Line 68: In the introduction, the authors highlight that a lower infection dose was required to establish infection. In figure 2, however, they do not state how many of the mice per group were actually infected. Did the authors test this or monitor viral loads over time? Is the absence of weight loss in some animals due to absence of infection?

Similarly, the authors state that they observed "faster disease progression and severity after infection with a low dose of B.1.1.7" without demonstrating that the mice were actually infected.

3) Figure 4C, D; lines 169/170: The authors highlight a few antibodies that require 2- to 3-fold higher concentrations to inhibit the N501Y variant (compared to WT). According to the source files, however, there are also antibodies that require 2-fold lower concentrations to inhibit the N501Y or N439K variants. These findings are not mentioned in the text. Which of the antibodies tested show statistically significant differences?

4) Figure 4C, D: About half of the antibodies do not reach an inhibition of 50% in the concentrations tested. Still, the authors set the respective IC50 values to 100 µg/ml and included them in the correlation/regression analyses. This may be misleading.

5) To examine whether the affinity of N501Y and ACE2 is involved in the efficient establishment of infection and development of disease, the authors showed that the weight of transgenic humanized ACE2 mice infected with B.1.1.7 is more decreasing than that with Freiberg. However, in fact, it is difficult to determine whether the virus is really high based on this data alone. The authors should show that the virus titer in the lung of B.1.1.7 or Freiberg infected mice and the number of SARS-CoV-2 positive cells in the infected mice lung using immunohistochemistry. Moreover, the causes of mice weight loss induced by virus infection is various. The authors should describe the amino acid difference between Freiberg isolation and B.1.1.7 strain or each strain's accession number. If there are amino acid mutation in the virus protein involved in pathogenicity, the authors describe it in Discussion section. It is also strange that no weight loss was observed in Freiberg strain-infected mice, as a cited paper demonstrated sever weight loss of SARS-CoV-2 strain 2019n CoV/USA_WA1/2020infected mice (more than UK strain in this study actually).

6) In Figure 1, the authors examined the effect of N439K and N501Y in biophysical interaction between S and ACE2 proteins. These data suggest that the mutations increase the affinity with ACE2. In Figure 2, to examine whether the increased affinity of the N501Y variants for ACE2 was associated with a more efficient establishment of infection and development of disease, the authors challenged K18-hACE2 mice with clinically isolated two viruses including B.1.1.7 strain. However, since each clinical isolate contains about 30 mutations, it is not suitable as a model for examining the effects of specific mutations such as N501Y. In several reports, reverse genetics system has already been established. To clarify the role of mutations in viral infection or disease development, recombinant virus with specific mutation is suitable as the model.

7) Several reports suggest that mutations in S proteins support the infection of SARS-CoV-2 into mouse cells. It is possible that B.1.1.7 can infect to the cells through the interaction with mouse ACE2, therefore the authors should check the infectivity of B.1.1.7 to WT mouse.

8) To claim that analysis of convalescent sera using a PRNT virus neutralization platform showed no difference in the neutralization efficacy against the WT and B1.1.7 strains, the authors should show the neutralization efficacy against the B.1.351 or P.1 as a positive control.*Reviewer #1:*

In their present manuscript, Bayarri-Olmos and colleagues investigate the effects of SARS-CoV-2 Spike mutations N501Y and N439K on ACE2 binding, viral pathogenicity in mice and/or antibody neutralization. Using biolayer interferometry, they convincingly show that the N501Y change increases affinity to the receptor-binding domain of human ACE2 by about 8-fold. Furthermore, transgenic mice infected with the B.1.1.7 variant (harboring N501Y) show a more pronounced weight loss at 6 and 7 days post infection than mice infected with a control virus (Freiburg isolate FR-4286). While interaction of ACE2 with the N501Y RBD was slightly less sensitive to convalescent sera than the respective WT RBD control, the authors observed no significant difference in a viral neutralization assay using B.1.1.7 and FR-4286.

One major strength of the manuscript is the combination of several independent approaches to elucidate the impact of the N501Y and/or N439K mutation on ACE2 binding, pathogenicity and antibody evasion. However, some of the main conclusions are not fully justified by the data shown:

1) The exact sequence of the control virus (FR-4286) remains unclear. Thus, it remains unclear whether the N501Y mutation or any other changes are responsible for the observed differences in weight loss in infected mice (Figure 2).

2) Although the authors do not confirm successful infection of their ACE2 transgenic mice, they conclude that a lower infection dose was required to establish infection (Figure 2; line 68).

3) In the abstract, the authors highlight that "the 501Y variant showed a minor, but significant elevated evasion potential of RBD:ACE-2 antibody neutralization" when "challenged with […] monoclonal antibodies". While this may be true for some antibodies, others showed the opposite effect in their inhibition assay (Figure 4C/D). Furthermore, they do not observe any differences in neutralization sensitivity when performing a (more relevant) plaque reduction neutralization test.*Reviewer #2:*

The author's aim is to reveal the functional biophysical characterization of N501Y mutation in the receptor binding domain (RBD) of SARS-CoV-2 spike protein. First, they showed that the interaction of recombinant RBD N501Y and ACE2 is higher affinity than that of RBD WT.

This result indicated the possibility of the B.1.1.7 strain having N501Y mutation is more efficient infection than SARS-CoV-2 Freiberg isolate. In fact, they showed that B.1.1.7 strain induced weight reduction in transgenic humanized ACE2 mice at lower doses than Freiberg　isolate. Moreover, this paper investigated the relationship of N501Y and immune evasion. They showed that the antibody capacity to neutralize N501Y and ACE2 interaction is decreased using the sera of convalescent individuals and monoclonal antibodies.

However, it has been already proved from the three-dimensional structural analysis that the affinity of N501Y mutant was increased toward hACE2, while it seems first study proving it using a recombinant protein. Moreover, evasion of B.1.1.7 strain from vaccine-induced immunity has already been demonstrated. This study is potentially interesting, but the conclusion of this study is not novel and several experiments will be needed for elucidation of the exact effect of N501Y substitution.*Reviewer #3:*

'The B.1.1.7 SARS-CoV-2 variant exhibits significantly higher affinity for ACE-2 and accelerates disease severity in transgenic hACE-2 mice' by Bayarri-Olmos et al., examines the impact of the N501Y mutation on binding to the human ACE2 protein and further explores the virulence of the B.1.1.7 variant of concern using the transgenic k18 hACE2 mouse model. Using a combination of mechanistic in vitro techniques, classical neutralization assays and in vivo mouse infections the authors found subtle but significant differences between peptides and/or viruses containing the N501Y mutation vs. wild type controls. These findings are generally convincing but particularly in the case of the mouse model data, require more detail explain the B.1.1.7 phenotype.

The B.1.1.7 variant was first recognized in late 2020 and rapidly became the dominant circulating strain in most countries. While epidemiological modeling clearly showed a transmission advantage, and in silico studies suggested increase affinity to hACE2, full details to explain the rise of B.1.17 have been lacking. In this manuscript Bayarri-Olmos et al., examine the role of the N501Y mutation in hACE2 binding as well as the broader question of the VOC B.1.1.7 virulence using the k18 mouse model. While much has recently been written about B.1.1.7 and other variants of concern, detailed mechanistic wet lab studies are lacking and this work is a welcome addition to the field.

The authors demonstrate that spike RBDs containing the N501Y mutation had similar stability as wild type RBDs but had 8 fold higher affinity for hACE2. This precise biophysical characterization is very convincing and provides important mechanistic insight into the spread of B.1.1.7; however; the RBD domain exists as part of a larger protein (and that as part of a trimer) and changes in quaternary structure or protein stability may impact the assays used in this work.

in vivo infection using the k18 mouse model shows more weight loss after B.1.1.7 infection than after infection with an early isolate that does not contain the N501Y mutation, however it is unclear if there is any difference in virus replication or spread to neurotropism between the two strains. The weight loss data presented shows a clear difference in the lower dose, however further details l to address the mechanism of increased pathogenesis caused by B.1.1.7 infection is important to explain this finding.

Consistent with other reports, Byarri-Olmos et al., found similar virus neutralization of B.1.1.7 and a non N501Y control using convalescent serum. Use of a live virus assay instead of a pseudovirus assay in this report is a strength.

Overall, the data presented support the author's conclusion that the change in Spike-ACE2 binding affinity caused by the N501Y mutation is responsible for the increased transmission of B.1.1.7. The biophysical data presented in this work provides an important mechanism behind enhanced transmission of B.1.1.7 around the world and adds important understanding to the evolving COVID-19 pandemic.

[Editors’ note: further revisions were suggested prior to acceptance, as described below.]

Thank you for resubmitting your work entitled "The α/B.1.1.7 SARS-CoV-2 variant exhibits significantly higher affinity for ACE-2 and requires lower inoculation doses to cause disease in K18-hACE2 mice" for further consideration by *eLife*. Your revised article has been evaluated by Miles Davenport (Senior Editor) and a Reviewing Editor.

The manuscript has been improved but there are some remaining issues that need to be addressed, as outlined below:

1. In the revised manuscript, the authors provided the data about viral load in the lungs of mice infected with SARS-CoV-2 only at day 2 after infection. The authors should show the data at multiple time points, such as day 1 and day 4. In addition, to investigate the pathogenicity, pathological analysis should also be performed. Furthermore, the source file is incorrect since identical values are provided for weight and viral load.

2. The authors should reply to the reviewer's comment: "Is is also strange that no weight loss was observed in Freiberg strain-infected mice, as a cited paper demonstrated severe weight loss of SARS-CoV-2 strain 2019n CoV/USA_WA1/2020 infected mice".

3. In several reports, a simple reverse genetics method has been reported. The authors should make mutant recombinant SARS-CoV-2 carrying N439K and/or N501Y by reverse genetics and examine the characteristics of mutant recombinant viruses.

4. Figure 4: While the authors excluded all mAbs with IC50s above the concentrations tested, they still provide IC50 values of 100 µg/ml in the respective source file. This may be misleading.

5. Since the authors have sequenced the Freiburg isolate, the respective sequence may be deposited and an accession number should be provided.

6. It seems that a preliminary version of the rebuttal was submitted. Particularly the introduction requires an update of the literature. For example, B1.621 may be introduced as "Mu variant of interest" (line 55) and P.3/Theta (line 54) is no longer considered a variant of interest by the WHO. Furthermore, some of the preprints cited have been published in the meantime (see e.g. line 239).

7. Figure 4C, D: The authors state that 1.5- to 1.8-fold effects "may be caused by biological/technical variation". Nevertheless, their values are based on a single duplicate measurement only. I strongly recommend performing three independent experiments since this is a key experiment of the study.

8. Lines 76-79: "This was associated with a lower infection dose requirement to establish infection in transgenic humanized ACE-2 mice challenged with the α/B.1.1.7 variant compared to an early 2020 isolate and resulted in increased disease severity." Since the previous sentence mentions both, the N439K and N501Y change, this statement may be misleading, given that the two isolates used do not differ at Spike position 439.

9. Line 386: Typo: the original amino acid residue at position 982 is missing.

As mentioned above, the reviewers think the manuscript was improved. However, the reviewers also think more in vivo data, such as the pathological analysis and experiments using recombinant virus, should be included. Addressing these issues would be essential for the acceptance.

---

## [Author Response]

Essential revisions:1) To assess the pathogenicity and potential immune evasion of B.1.1.7, the authors use a control isolate termed FR-4286. In which sites do the Spike proteins of B.1.1.7 and FR-4286 differ? The respective sequences may be provided. Can the authors exclude mutations that occurred during propagation in cell culture (in Vero cells)?

Sequence analysis of the variant FR-4286 shows that it is the Wuhan-like early European B.1 lineage strain, with the lineage characterizing mutations S:D614G, ORF1b:P314L. In addition, the strain carries an E:L37R amino acid variation.

This means, that the spike proteins of B.1.1.7 strain differ from FR-4286 on position S:N501Y, S:A570D, S:T716I, S:P681H, S:982A, S:D1118H, S:del69/70, S:del144/145. Both the B.1.1.7 and the FR-4286 strains contains the S:D614G and ORF1b:P314L mutations. This is based on sequencing of the isolates in Vero cells. This information has been added to the methods section in the revised manuscript (pp. 18-19, lines 347-366).

2) Line 68: In the introduction, the authors highlight that a lower infection dose was required to establish infection. In figure 2, however, they do not state how many of the mice per group were actually infected. Did the authors test this or monitor viral loads over time? Is the absence of weight loss in some animals due to absence of infection? Similarly, the authors state that they observed "faster disease progression and severity after infection with a low dose of B.1.1.7" without demonstrating that the mice were actually infected.

In the revised manuscript we now provide data on viral load in the lungs of mice infected with FR-4286 vs. Α (B.1.1.7) at day 2 after infection (New figure 2B). These data confirm that identical infection doses of Α and FR-4286 leads to higher viral load of the Α variant. In the graph we show values from individual mice, which demonstrate that the mice were actually infected.

3) Figure 4C, D; lines 169/170: The authors highlight a few antibodies that require 2- to 3-fold higher concentrations to inhibit the N501Y variant (compared to WT). According to the source files, however, there are also antibodies that require 2-fold lower concentrations to inhibit the N501Y or N439K variants. These findings are not mentioned in the text. Which of the antibodies tested show statistically significant differences?

We highlighted the antibodies that were compromised by the single mutations, for we considered those were the most biologically relevant. The editors are right when they mentioned that two antibodies neutralize the variants more effectively (1.5- and 1.8-fold), even when they were generated after RBD wt immunization. Nonetheless, such small differences may be caused by biological/technical variation.

We have updated the figures explicitly stating the fold difference (both decreased and enhanced neutralization) for all antibodies when it was above 1.5-fold. We have also updated the text in the Figure legend (L-196–198) and Results (L. 176–185). We did not perform statistical analyses on the individual antibodies, because the kDs were interpolated from duplicate measurements.

4) Figure 4C, D: About half of the antibodies do not reach an inhibition of 50% in the concentrations tested. Still, the authors set the respective IC50 values to 100 µg/ml and included them in the correlation/regression analyses. This may be misleading.

We understand the concerns raised regarding including non-inhibitory antibodies in the statistical analyses. We considered relevant to include all data in the analysis. Notwithstanding, we have now excluded from the analyses all mAbs with IC50s above the concentrations tested (previously normalized to 100). Figure 4C, D have been updated accordingly, as well as the figure legend (196–198).

5) To examine whether the affinity of N501Y and ACE2 is involved in the efficient establishment of infection and development of disease, the authors showed that the weight of transgenic humanized ACE2 mice infected with B.1.1.7 is more decreasing than that with Freiberg. However, in fact, it is difficult to determine whether the virus is really high based on this data alone. The authors should show that the virus titer in the lung of B.1.1.7 or Freiberg infected mice and the number of SARS-CoV-2 positive cells in the infected mice lung using immunohistochemistry. Moreover, the causes of mice weight loss induced by virus infection is various. The authors should describe the amino acid difference between Freiberg isolation and B.1.1.7 strain or each strain's accession number. If there are amino acid mutation in the virus protein involved in pathogenicity, the authors describe it in Discussion section. It is also strange that no weight loss was observed in Freiberg strain-infected mice, as a cited paper demonstrated sever weight loss of SARS-CoV-2 strain 2019n CoV/USA_WA1/2020infected mice (more than UK strain in this study actually).

Please see response to point 1 and 2 above, where we address the issues on mutations in the spike protein and mutations virus load in individual mice.

6) In Figure 1, the authors examined the effect of N439K and N501Y in biophysical interaction between S and ACE2 proteins. These data suggest that the mutations increase the affinity with ACE2. In Figure 2, to examine whether the increased affinity of the N501Y variants for ACE2 was associated with a more efficient establishment of infection and development of disease, the authors challenged K18-hACE2 mice with clinically isolated two viruses including B.1.1.7 strain. However, since each clinical isolate contains about 30 mutations, it is not suitable as a model for examining the effects of specific mutations such as N501Y. In several reports, reverse genetics system has already been established. To clarify the role of mutations in viral infection or disease development, recombinant virus with specific mutation is suitable as the model.

We agree that the RBD N501Y variant does not directly reflect the full spike/virus infection situation in vivo. For the biophysical characterization, however, we considered that side-by-side comparisons of RBD harboring single mutations is a appropriate approach. The RBD determines the affinity towards ACE-2, and while we cannot discard minor effects from mutations outside the RBD, it is likely that the RBD_N501Y/ACE-2 interaction can be extrapolated to the B.1.1.7 spike. Nonetheless, we do not claim that the RBD affinity increase caused by the N501Y is the sole reason for the apparent higher transmission and disease progression in vivo. We understand the concern and have introduced a paragraph in the discussion regarding the other mutations in the spike gene.

7) Several reports suggest that mutations in S proteins support the infection of SARS-CoV-2 into mouse cells. It is possible that B.1.1.7 can infect to the cells through the interaction with mouse ACE2, therefore the authors should check the infectivity of B.1.1.7 to WT mouse.

We agree that this is an important point. We have not been able to do the infection studies in wt mice but there is a recent report suggesting that B.1.1.7 can infect mice cells to some degree, though not as efficient as the P.1 and B.1.351. (https://doi.org/10.1101/2021.03.18.436013). We have included a paragraph in the discussion highlighting this.

8) To claim that analysis of convalescent sera using a PRNT virus neutralization platform showed no difference in the neutralization efficacy against the WT and B1.1.7 strains, the authors should show the neutralization efficacy against the B.1.351 or P.1 as a positive control.

Our ELISA-based neutralization assay highly correlates with the PRNT, as shown in the figure from Bayarri-Olmos et al., (The Journal of Immunology, 2021, in print).

Using our assay in a publication currently under review in another journal, we have compared the neutralization efficacy of convalescent sera against RBD harboring different mutations, like the P.1. A modified figure is included, see Author response image 1.

**Author response image 1. sa2fig1:** Inhibitory potency of COVID-19 convalescent patient sera against RBD variants. Antibody-mediated inhibition of serum from recovered COVID-19 patients (*n* = 150) against RBD wt, N501Y and P.1 (harboring the Y417N, E484K, and N501Y mutations). Statistical comparisons between groups were performed using the Friedman test with Dunn’s multiple comparisons. Orange lines represent medians.

All in all, the PRNT is a robust and well-characterized assay, and the moderate decrease in neutralization by the N501Y variant is in accordance with a growing body of evidence.

[Editors' note: further revisions were suggested prior to acceptance, as described below.]

The manuscript has been improved but there are some remaining issues that need to be addressed, as outlined below:1. In the revised manuscript, the authors provided the data about viral load in the lungs of mice infected with SARS-CoV-2 only at day 2 after infection. The authors should show the data at multiple time points, such as day 1 and day 4. In addition, to investigate the pathogenicity, pathological analysis should also be performed. Furthermore, the source file is incorrect since identical values are provided for weight and viral load.

Thank you for the relevant suggestion. Generally, we and others (e.g. ref 32) do not see profound differences in the titer at day 2-4 p.i. and around day 6-7 the virus is being eliminated from the lungs in this model. We believe that day 2 p.i is best representing the disease development and monitoring the viral load at day 1 and 4, although adding to the breadth of the analysis, would not add significant additional information. We have done TCID50 assays on lung tissues from the in vivo experiments that essentially resemble the viral load measurements (from figure 2b) but with a different observed statical significance. The results from the TCID50 are presented in Author response image 2. These data can be included in the manuscript if the editors think it will be of value.

Thank you for noticing that the source files for figure 2b were not uploaded correctly. These are now corrected and attached in the resubmission.

2. The authors should reply to the reviewer's comment: "Is is also strange that no weight loss was observed in Freiberg strain-infected mice, as a cited paper demonstrated severe weight loss of SARS-CoV-2 strain 2019n CoV/USA_WA1/2020 infected mice".

The authors agree that there are differences in disease development in K18-hACE mice infected with the parental virus when compared to other published data. However, as also reported in the literature, and similar to humans, the mice exhibit age-dependence development of disease. The mice used in this study were in the lower age-range, and hence likely explaining why only limited development of disease was observed. Interestingly, the mice infected with the B.1.1.7 variant did nevertheless develop severe disease and higher viral load despite the low age of the mice.

3. In several reports, a simple reverse genetics method has been reported. The authors should make mutant recombinant SARS-CoV-2 carrying N439K and/or N501Y by reverse genetics and examine the characteristics of mutant recombinant viruses.

We fully agree with the editors and think that this is an interesting suggestion. However due to safety restrictions in working with genetically modified SARS-CoV2 this is currently not an option for us within the Danish legislation. This would require a combined GMO-3 and BSL3 facility, which currently does not exist in Denmark.

4. Figure 4: While the authors excluded all mAbs with IC50s above the concentrations tested, they still provide IC50 values of 100 µg/ml in the respective source file. This may be misleading.

We have updated the source file and highlighted the mAbs with IC50s above the range of concentrations tested as “excluded”. We have given them an arbitrary value and excluded them from any analyses.

5. Since the authors have sequenced the Freiburg isolate, the respective sequence may be deposited and an accession number should be provided.

The Freiburg isolate sequence has been uploaded and has the accession ID no: EPI_ISL_852748. This has been disclosed and included in the revised manuscript.

6. It seems that a preliminary version of the rebuttal was submitted. Particularly the introduction requires an update of the literature. For example, B1.621 may be introduced as "Mu variant of interest" (line 55) and P.3/Theta (line 54) is no longer considered a variant of interest by the WHO. Furthermore, some of the preprints cited have been published in the meantime (see e.g. line 239).

We thank the editors for highlighting this. We have updated the introduction regarding the Mu VOI and the theta, a former VOI according to https://www.who.int/en/activities/tracking-SARS-CoV-2-variants/. We have also revised the reference list and updated all articles that have been published on the meantime.

7. Figure 4C, D: The authors state that 1.5- to 1.8-fold effects "may be caused by biological/technical variation". Nevertheless, their values are based on a single duplicate measurement only. I strongly recommend performing three independent experiments since this is a key experiment of the study.

We agree. And as suggested, we have performed the experiment again, as three independent replicates. The figure and source file have been updated accordingly.

8. Lines 76-79: "This was associated with a lower infection dose requirement to establish infection in transgenic humanized ACE-2 mice challenged with the α/B.1.1.7 variant compared to an early 2020 isolate and resulted in increased disease severity." Since the previous sentence mentions both, the N439K and N501Y change, this statement may be misleading, given that the two isolates used do not differ at Spike position 439.

We agree that the statement could be misleading and have changed the paragraph in the manuscript.

9. Line 386: Typo: the original amino acid residue at position 982 is missing.

We apologize for the oversight. It has been corrected to S:S982A.